# Delta-CoMe: Training-Free Delta-Compression with Mixed-Precision for Large Language Models

**Bowen Ping**[1]* **Shuo Wang**[2]* **Hanqing Wang**[3] **Xu Han**[2,4,5] **Yuzhuang Xu**[2] **Yukun Yan**[2]
**Yun Chen**[3] **Baobao Chang**[1] **Zhiyuan Liu**[2,4,5]† **Maosong Sun**[2,4,5]†

[1]Peking University   [2]Dept. of Comp. Sci. & Tech., Tsinghua University, Beijing, China
[3]Shanghai University of Finance and Economics
[4]Institute for AI, Tsinghua University, Beijing, China
[5]Beijing National Research Center for Information Science and Technology

## Abstract

Fine-tuning is a crucial process for adapting large language models (LLMs) to diverse applications. In certain scenarios, such as multi-tenant serving, deploying multiple LLMs becomes necessary to meet complex demands. Recent studies suggest decomposing a fine-tuned LLM into a base model and corresponding delta weights, which are then compressed using low-rank or low-bit approaches to reduce costs. In this work, we observe that existing low-rank and low-bit compression methods can significantly harm the model performance for task-specific fine-tuned LLMs (e.g., WizardMath for math problems). Motivated by the long-tail distribution of singular values in the delta weights, we propose a delta quantization approach using mixed-precision. This method employs higher-bit representation for singular vectors corresponding to larger singular values. We evaluate our approach on various fine-tuned LLMs, including math LLMs, code LLMs, chat LLMs, and even VLMs. Experimental results demonstrate that our approach performs comparably to full fine-tuned LLMs, surpassing both low-rank and low-bit baselines by a considerable margin. Additionally, we show that our method is compatible with various backbone LLMs, such as Llama-2, Llama-3, and Mistral, highlighting its generalizability. [3]

## 1 Introduction

Large language models (LLMs) (Touvron et al., 2023; Jiang et al., 2023) are increasingly becoming the standard for a wide range of downstream tasks (Luo et al., 2023a; Yu et al., 2023; Wei et al., 2023; Luo et al., 2023b; Liu et al., 2024a; Wang et al., 2023), significantly surpassing conventional small models. To meet the demands of various application domains and scenarios, many researchers direct their attention to developing advanced alignment or adaptation algorithms together with diverse training data to learn aligned LLMs based on generally pre-trained models. For instance, Luo et al. (2023a) propose a reinforcement learning from evol-instruct feedback (RLEIF) method to construct LLMs with strong mathematical reasoning abilities. Similarly, Yu et al. (2023) employ a bootstrapping method to diversify mathematical questions and then fine-tune open-source LLMs to build mathematical models. For code generation, Luo et al. (2023b) adapt the evol-instruct method to the coding domain, resulting in the WIZARDCODER model, which demonstrates superior coding abilities compared to generally trained LLMs. Additionally, Wei et al. (2023) enhance the capabilities of open-source code LLMs by using automatically generated high-quality instruction data based on

---

* Equal contribution.
† Corresponding authors.
[3] Code will be publicly available at `https://github.com/thunlp/Delta-CoMe`.

38th Conference on Neural Information Processing Systems (NeurIPS 2024).

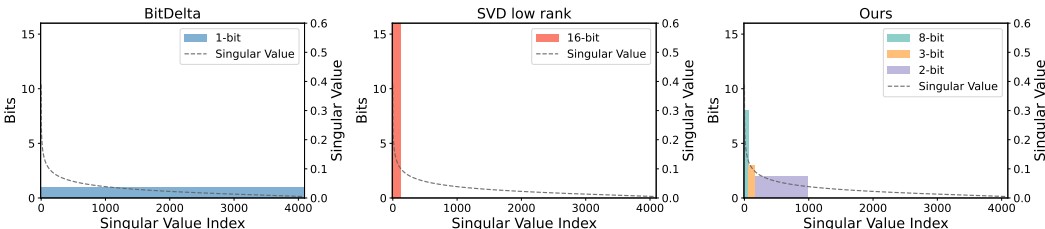

Figure 1: **Left**: illustration of BitDelta (Liu et al., 2024b), which employs 1-bit quantization for all the delta weights. **Middle**: illustration of low-rank compression (Ryu et al., 2023b), retaining the top-$k$ singular values and the corresponding singular vectors. **Right**: illustration of the proposed Delta-CoMe method, which represents the singular vectors of larger singular values using high-bit vectors while compressing the singular vectors of smaller singular values into low-bit representations. This method is inspired by the long-tail distribution of singular values in delta weights.

existing code snippets. Wang et al. (2023) utilize various resources of mixed quality and design a new conditioned reinforcement learning fine-tuning method to train the OPENCHAT model. Beyond the text modality, some studies propose fine-tuning pre-trained LLMs to understand other modalities. For instance, Liu et al. (2024a) construct a multi-modal instruction tuning dataset and develop the LLAVA model, which can understand both text and images.

Building on the aforementioned alignment approaches, LLMs are endowed with specialized capabilities that align with distinct user demands and application requirements (Liu et al., 2024b). In certain scenarios, deploying multiple LLMs with different abilities is necessary. For example, in multi-tenant serving, different LLMs may be needed to satisfy various users. Additionally, some complex tasks consist of multiple sub-tasks, each requiring different model capabilities. To address these tasks, we should organize and deploy a group of LLMs simultaneously. A straightforward question arises: why not use a single general LLM that encompasses all the necessary capabilities? For example, we could develop one model that can both understand images and generate code programs. To our knowledge, LLMs with various capabilities (e.g., GPT-4[4]) typically have an enormous number of parameters, making them impractical for resource-limited situations (e.g., edge-side scenarios).

In pursuit of this objective, a field of research advocates for the minimization of expenses associated with multi-model serving. Delta-compression emerges as a crucial and viable approach in this context, offering the potential to decrease both storage requirements and GPU memory utilization in scenarios involving multiple models. The primary objective of delta-compression is to minimize the size of the delta weights between aligned and pre-trained LLMs (e.g., LLAMA-2-CHAT and LLAMA-2). Ryu et al. (2023b) identify the low-rank nature of delta weights and enhance storage efficiency through low-rank approximation. Alternatively, Liu et al. (2024b) propose a 1-bit quantization approach, termed BitDelta, to further reduce the size of delta weights. They validate the effectiveness of BitDelta across various chat models, including LLAMA-2-CHAT (Touvron et al., 2023), VICUNA[5], and WIZARDLM (Xu et al., 2023). In this work, we reassess the performance of both low-rank and low-bit delta-compression methods across a diverse range of aligned LLMs, encompassing mathematical, coding, chat, and multi-modal LLMs. Our experimental results (e.g., Table 3) reveal that current low-rank and low-bit compression techniques may significantly degrade the performance of aligned LLMs. These results motivate us to explore more advanced delta-compression methods capable of achieving performance nearly equivalent to the aligned LLMs before compression.

Inspired by the long-tail distribution of singular values, as illustrated in Figure 1, we propose allocating higher-bit representations for singular vectors associated with larger singular values, given their greater impact on the approximation of delta weights prior to compression. Conversely, for singular vectors associated with smaller singular values, we employ low-bit formats to reduce the delta size. For singular values that are extremely small, we omit the corresponding singular vectors altogether. The resulting method, which we term Delta-CoMe, can be viewed as a hybrid of low-rank and low-bit compression techniques. Delta-CoMe outperforms both the low-rank compression method and BitDelta. Moreover, our method achieves performance comparable to that of the full

---

[4] https://chatgpt.com
[5] https://lmsys.org/blog/2023-03-30-vicuna

aligned LLMs. For instance, Delta-CoMe attains an average score of 53.2 across eight representative tasks, closely matching the average score of 53.5 achieved by the aligned LLMs. In comparison, the scores of the low-rank and low-bit baselines are 47.8 and 49.3, respectively.

Further, we compare the performance of the involved delta-compression methods to LoRA (Hu et al., 2022), a widely-used delta-tuning approach (Wang et al., 2024). The primary distinction between delta-compression and delta-tuning is that delta-compression first optimizes the full model and then converts the modified weights into a lightweight module, reducing inference costs in multi-model settings. In contrast, delta-tuning primarily aims to lower training costs. Our experimental results demonstrate that the proposed Delta-CoMe method significantly outperforms LoRA, with scores of 41.9 versus 29.8, respectively. These results suggest that delta-compression can deliver superior performance in multi-model settings compared to delta-tuning.

Finally, Delta-CoMe can achieve more than $10\times$ GPU memory and disk storage savings, enabling the deployment of multiple models with limited resources. For practical application, we implement a Triton (Tillet et al., 2019) kernel tailed for Delta-Come, achieving approximately a $3\times$ speedup compared to the PyTorch implementation.

Our contribution can summarized as follows:

- We propose a mixed-precision delta-compression method that employs varying bit-widths for different singular vectors based on their singular values;

- We validate the effectiveness of the proposed method across different types of aligned LLMs of varying sizes, including mathematical, coding, chat, and multi-modal LLMs;

- We conduct in-depth analyses to understand the superior performance of our method over low-rank and low-bit baselines. Our method can also outperform delta-tuning approaches such as LoRA, demonstrating that the proposed delta-compression method is more practical for multi-model serving scenarios.

- We verify that the proposed method can achieve over $10\times$ saving in GPU memory and disk storage. By constructing a Triton kernel, we can achieve approximately a $3\times$ speedup, demonstrating the hardware compatibility of Delta-CoMe.

## 2 Related Work

### 2.1 Delta-Compression

Recently, delta-compression has garnered increasing interest in the LLM community due to its ability to substantially diminish the storage and inference expenses associated with serving multiple models. GPT-Zip extends the GPTQ approach (Frantar et al., 2023) to compress the delta weights between aligned models and the backbone model, successfully using 2-bit delta weights to approximate the model. Additionally, they sparsify the quantized delta weights to further reduce storage costs. However, the sparsification technique can hardly reduce GPU memory usage during inference. Similarly, Yu et al. (2024) find that dropping the majority of the delta weights has a limited effect on the performance of aligned LLMs. Ryu et al. (2023a) identify the low-rank property of delta weights and propose reducing the storage requirements of aligned LLMs through low-rank approximation. Yao & Klimovic (2023) adopt the concept of delta-compression to develop a multi-tenant serving system, DeltaZip. Most recently, Liu et al. (2024b) introduced BitDelta, which successfully quantizes the delta weights into 1-bit. However, they only examined the performance of this compression method using chat LLMs, leaving a wide range of other types of aligned LLMs unexplored. In this work, we propose leveraging the benefits of both low-rank and low-bit compression methods by using varying bit-widths to represent different components of the delta weights. We evaluate representative low-rank and low-bit delta-compression methods across various types of aligned LLMs to provide a comprehensive comparison of these methods.

### 2.2 Model Compression with Mix-Precision

Using mixed-precision to compress the model weights is an effective technique that has been investigated in many previous studies. SpQR (Dettmers et al., 2023) isolates a small number of outlier weights and retains them with high-precision, while keeping the other weights at low-precision,

resulting in a significant performance improvement. Based on activations, Agile-Quant (Shen et al., 2024) utilizes token pruning to achieve mixed-precision quantization of both weights and activations. Bablani et al. (2023) propose employing varying bit-widths for different layers of the model, while Yao et al. (2021) propose quantizing activations and model weights with different precisions. In this work, we propose using mixed-precision compression for different singular vectors of the delta model, marking the first method to introduce mixed-precision compression for delta weights.

## 3 Approach

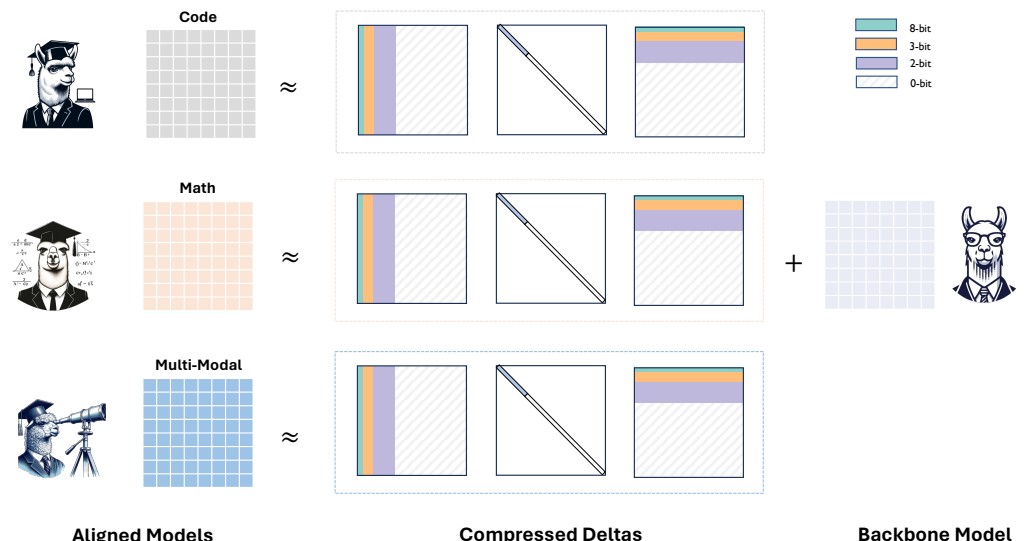

Figure 2: Illustration of Delta-CoMe, where we utilize varying bit-widths for singular vectors with different singular values. Singular vectors corresponding to larger singular values are assigned higher bit-widths. For extremely small singular values, we omit the singular vectors (i.e., 0-bit).

### 3.1 Preliminaries

For a **backbone LLM** $\boldsymbol{\theta}_b$, we can customize it into an **aligned model** $\boldsymbol{\theta}_a$ for a specific purpose using advanced alignment algorithms (Xu et al., 2023; Luo et al., 2023a; Yu et al., 2023; Luo et al., 2023b; Wei et al., 2023; Liu et al., 2024a). In some practical scenarios, as mentioned in Section 1, we may need to deploy multiple LLMs at the same time. Formally, we should store and deploy a series of aligned LLMs $\left\{ \boldsymbol{\theta}_a^{(1)}, \cdots, \boldsymbol{\theta}_a^{(N)} \right\}$, where $N$ is the number of aligned models. The total size of the group of aligned models is $N \times M$, where $M$ is the size of one model. We use $\boldsymbol{\Delta}$ to represent the delta weights between the aligned model and the backbone model, which is given by

$$\boldsymbol{\Delta}^{(n)} = \boldsymbol{\theta}_a^{(n)} - \boldsymbol{\theta}_b, \tag{1}$$

where $\boldsymbol{\theta}^{(n)}$ is the $n$-th aligned LLM. Note that the sizes of $\boldsymbol{\Delta}^{(n)}$, $\boldsymbol{\theta}_a^{(n)}$, and $\boldsymbol{\theta}_b$ are the same.

Delta-compression aims to compress the delta weights $\boldsymbol{\Delta}^{(n)}$ into $\hat{\boldsymbol{\Delta}}^{(n)}$, where the latter has significantly fewer parameters. After delta-compression, we can only maintain one backbone model and $N$ compressed delta models: $\left\{ \boldsymbol{\theta}_b, \hat{\boldsymbol{\Delta}}^{(1)}, \cdots, \hat{\boldsymbol{\Delta}}^{(N)} \right\}$. The total size is decreased from $N \times M$ to $(1 + \alpha N) \times M$, where $\alpha$ is the compression ratio. During inference, we can restore each aligned LLM in the following way:

$$\hat{\boldsymbol{\theta}}_a^{(n)} = \boldsymbol{\theta}_b + \hat{\boldsymbol{\Delta}}^{(n)}. \tag{2}$$

For a good delta-compression method, we expect it can achieve a smaller $\alpha$, while making $\hat{\boldsymbol{\theta}}_a^{(n)}$ attain comparable performance with $\boldsymbol{\theta}_a^{(n)}$. BitDelta (Liu et al., 2024b), to our knowledge, is the most recent study that successfully quantizes delta weights into 1-bit, which means that $\alpha = 1/16$ when the

original aligned model is represented by FP16 or BF16. In this work, we propose to improve the performance of delta-compression methods by inducing mixed-precision quantization, which will be detailed in the following sub-sections.

## 3.2 Delta Decomposition

Previous works have investigated mixed-precision model compression methods at the token (Shen et al., 2024) or layer level (Bablani et al., 2023). For delta-compression, we propose employing mixed-precision for different singular vectors. We first use the SVD algorithm to decompose each delta matrix:

$$\Delta \mathbf{W} = \mathbf{U}\mathbf{\Sigma}\mathbf{V}^\top, \tag{3}$$

where $\Delta \mathbf{W} \in \mathbb{R}^{h_{\text{out}} \times h_{\text{in}}}$, $\mathbf{U} \in \mathbb{R}^{h_{\text{out}} \times h_{\text{out}}}$, $\mathbf{\Sigma} \in \mathbb{R}^{h_{\text{out}} \times h_{\text{in}}}$, $\mathbf{V} \in \mathbb{R}^{h_{\text{in}} \times h_{\text{in}}}$. Intuitively, the singular vectors associated with larger singular values have a greater impact on the approximation of the delta matrix $\Delta \mathbf{W}$, we thus spend more bits for these vectors to reduce the quantization error.

## 3.3 Mixed-Precision Quantization

Some representative quantization methods, such as GPTQ (Frantar et al., 2023), aims to minimize the following objective:

$$\hat{\mathbf{W}} = \text{Quant}_k(\mathbf{W}, \mathbf{X}) = \underset{\hat{\mathbf{W}}}{\text{argmin}} \, ||\mathbf{W}\mathbf{X} - \hat{\mathbf{W}}\mathbf{X}||^2, \tag{4}$$

where $\mathbf{X} \in \mathbb{R}^{h_{\text{in}}}$ is the input to the parameter $\mathbf{W}$ and $\hat{\mathbf{W}}$ is the corresponding quantized parameter. We use $\text{Quant}_k$ to denote the $k$-bit quantization algorithm. In this work, we employ the widely-used GPTQ method with group_size $= 128$ for cases where $k > 1$, and BitDelta for 1-bit quantization. For a certain group of singular vectors, let $r_{\text{begin}}$ and $r_{\text{end}}$ represent the start and end indices, respectively. The quantization of the singular vectors can be given by

$$\hat{\mathbf{V}}[:, r_{\text{begin}} : r_{\text{end}}]^\top = \text{Quant}_k(\mathbf{V}[:, r_{\text{begin}} : r_{\text{end}}]^\top, \mathbf{X}),$$
$$\hat{\mathbf{U}}[:, r_{\text{begin}} : r_{\text{end}}] = \tag{5}$$
$$\text{Quant}_k(\mathbf{U}[:, r_{\text{begin}} : r_{\text{end}}], \mathbf{\Sigma}[r_{\text{begin}} : r_{\text{end}}, r_{\text{begin}} : r_{\text{end}}]\hat{\mathbf{V}}[:, r_{\text{begin}} : r_{\text{end}}]^\top \mathbf{X}).$$

As illustrated in Figure 2, we use varying quantization bits for different groups of singular vectors. By employing different mixed-precision strategies, we can control the trade-off between achieving a small delta size and maintaining high performance. We will provide more details about the exploration of the mixing strategy in Section 5.1.

## 4 Experimental Setup

To thoroughly investigate the proposed delta-compression method Delta-CoMe and the involved baselines, we examine the performance of different methods across several tasks, which are typical applications of recent aligned LLMs.

### 4.1 Tasks

**Mathematical Problem Solving** Solving mathematical problems is a challenging task for modern LLMs. For this task, we employ GSM8K (Cobbe et al., 2021) and MATH (Hendrycks et al., 2021) as the evaluation datasets, which are among the most popular mathematical benchmarks for LLMs. The reported score is accuracy, which is estimated by comparing the ground-truth number with the result calculated by the model.

**Code Generation** The ability to process code is crucial for numerous practical applications, including data analysis and LLM-based agents. For this task, we use HumanEval (Chen et al., 2021) and MBPP (Austin et al., 2021) as the evaluation datasets, which are widely used in recent studies. The reported score is the pass rate, indicating that the model-generated code can successfully run the test cases in one pass (i.e., pass@1).

**Chat**   The chat ability enables LLMs to interact with users, providing helpful and safe suggestions or responses based on the user's requests. A good chat model is expected to be well aligned with human preferences. For evaluating chat LLMs, we select TruthfulQA (Lin et al., 2022) and SafetyBench (Zhang et al., 2023) as the evaluation datasets, which measure helpfulness and safety, respectively. The reported score is the accuracy, indicating that the choice of the model is correct.

**Multi-Modal Chat**   Vision-language models (VLMs) are attracting increasing attention due to their ability to process both text and images. Most recent VLMs are based on pre-trained visual encoders and language models, with the language models fine-tuned to understand the visual signal. For this task, we use GQA (Hudson & Manning, 2019) and TextVQA (Singh et al., 2019). The reported score is the accuracy, indicating that the choice of the model is correct.

### 4.2 Models

Table 1: Selected backbone and aligned models for the examined four tasks.

| Task | 7B Models | | 13B Models | |
|---|---|---|---|---|
| | Backbone | Aligned | Backbone | Aligned |
| Math | LLAMA-2 | WIZARDMATH-V1.0 | LLAMA-2 | WIZARDMATH-V1.0 |
| Code | CODELLAMA-PY | MAGICODERS-CL | CODELLAMA-PY | WIZARDCODER-PY-V1.0 |
| Chat | LLAMA-2 | LLAMA-2-CHAT | LLAMA-2 | LLAMA-2-CHAT |
| Multi-Modal | VICUNA-V1.5 | LLAVA-V1.5 | VICUNA-V1.5 | LLAVA-V1.5 |

For the four tasks, we provide the backbone and aligned models in Table 1. All the model weights are open-sourced by the authors. We use both 7B and 13B models to make a thorough comparison between different delta-compression models. During inference, we use greedy search.

### 4.3 Baselines

We employ two representative baselines, including SVD-based low-rank compression and Bit-Delta (Liu et al., 2024b). For the low-rank baseline, we re-implement the method, while for BitDelta, we use the code open-sourced by the authors.[6] All methods are evaluated on NVIDIA A100 GPUs.

## 5   Experimental Results

### 5.1   Exploration of Mixed-Precision Strategies

To determine which bit-width to use and how many singular vectors to quantize, we conduct a preliminary experiment using different mixed-precision strategies. We examine three types of strategies: single-precision, double-precision, and triple-precision settings. The size of the compressed delta remains consistent across all settings. For single-precision compression, we set $r_{\text{begin}}$ to 0, and $r_{\text{end}}$ is set to guarantee that the delta size is the same as BitDelta (Liu et al., 2024b). In other words, the compression ratio $\alpha$ for all settings is 1/16. Formally, for a delta matrix $\Delta\mathbf{W} \in \mathbb{R}^{h_{\text{out}} \times h_{\text{in}}}$, $r_{\text{begin}}$ and $r_{\text{end}}$ are set to satisfy the following equation:

$$k \times (r_{\text{end}} - r_{\text{begin}})(h_{\text{out}} + h_{\text{in}}) = 16 \times \alpha h_{\text{out}} h_{\text{in}}, \quad (6)$$

where $\alpha$ is set to 1/16 in our experiments, which is the same as BitDelta. In double-precision settings, $r_{\text{begin}}$ and $r_{\text{end}}$ are set to 0 and 2, respectively, for the first precision. For the second precision, $r_{\text{begin}}$

Table 2: Comparison of different mixed-precision strategies.

| # Precision | Setting | GSM8K |
|---|---|---|
| Single | 1 | 45.6 |
| | 2 | 50.6 |
| | **3** | **51.8** |
| | 4 | 51.6 |
| | 8 | 47.8 |
| | 16 | 43.3 |
| Double | 16 + 3 | 52.5 |
| | **8 + 3** | **53.1** |
| | 4 + 3 | 52.2 |
| | 3 + 2 | 52.3 |
| Triple | 16 + 8 + 3 | 53.2 |
| | 8 + 4 + 3 | 52.2 |
| | **8 + 3 + 2** | **53.6** |

---

[6]https://github.com/FasterDecoding/BitDelta.

is set to 2, and $r_{\text{end}}$ is adjusted so that the total delta size is 1/16 of the uncompressed delta. In triple-precision settings, $r_{\text{begin}}$ and $r_{\text{end}}$ are set to 0 and 2, respectively, for the first precision. $r_{\text{begin}}$ and $r_{\text{end}}$ are set to 2 and 34, respectively, for the second precision. For the third precision, $r_{\text{begin}}$ is set to 34, and $r_{\text{end}}$ is adjusted so that the total delta size is 1/16 of the uncompressed delta. Since the diagonal matrix $\Sigma$ occupies little storage, the averaged bit-width for triple-precision compression is approximately

$$\frac{h_{\text{out}} + h_{\text{in}}}{h_{\text{out}} h_{\text{in}}} \sum_{i=1}^{3} k^{(i)} (r_{\text{end}}^{(i)} - r_{\text{begin}}^{(i)}). \tag{7}$$

We conduct experiments on the math task, and the results are shown in Table 2. We find that the 3-bit setting performs best among the single-precision settings. Therefore, we keep the 3-bit setting and add other bit-widths to form double-precision settings. Among the double-precision settings, "8+3" achieves the highest score, which is then combined with an additional bit-width to form triple-precision settings. We find that the best double-precision setting can outperform the best single-precision setting, and the best triple-precision setting achieves the highest score across all the examined settings. We use "8+3+2" as the default setting in the following experiments.

## 5.2 Main Results

Tables 3 and 4 show the performance of different delta-compression methods on 7B and 13B models, respectively. Across all tasks, Delta-CoMe outperforms both baselines. While BitDelta (Liu et al., 2024b) can achieve near lossless performance on chat models, it significantly degrades the performance of math and code LLMs, a phenomenon not investigated by Liu et al. (2024b). Surprisingly, our method achieves good performance in the delta-compression of VLMs. To our knowledge, we are the first to investigate delta-compression for VLMs.

Table 3: The performance of different delta-compression methods on 7B aligned models.

| Method | $\alpha$ | WIZARDMATH | | MAGICODERS-CL | | LLAMA-2-CHAT | | LLAVA-V1.5 | | Ave. |
| --- | --- | --- | --- | --- | --- | --- | --- | --- | --- | --- |
| | | GSM8K | MATH | HumanEval | MBPP | SafetyBench | TruthfulQA | GQA | TextVQA | |
| Backbone | 1 | 11.0 | 2.9 | 38.4 | 47.6 | 41.7 | 38.9 | n/a | n/a | n/a |
| Aligned | 1 | 55.2 | 10.9 | 70.7 | 69.2 | 59.5 | 44.6 | 62.0 | 58.2 | 53.5 |
| Low-Rank | 1/16 | 43.2 | 8.0 | 56.7 | 65.7 | 55.4 | 42.5 | 57.7 | 53.3 | 47.8 |
| BitDelta | 1/16 | 45.6 | 8.6 | 57.3 | 65.9 | 59.3 | 41.1 | 59.7 | 56.9 | 49.3 |
| Delta-CoMe | 1/16 | **53.6** | **10.3** | **67.1** | **67.9** | **59.8** | **47.0** | **61.7** | **58.5** | **53.2** |

Table 4: The performance of different delta-compression methods on 13B aligned models.

| Method | $\alpha$ | WIZARDMATH | | WIZARDCODER | | LLAMA-2-CHAT | | LLAVA-V1.5 | | Ave. |
| --- | --- | --- | --- | --- | --- | --- | --- | --- | --- | --- |
| | | GSM8K | MATH | HumanEval | MBPP | SafetyBench | TruthfulQA | GQA | TextVQA | |
| Backbone | 1 | 17.8 | 3.9 | 43.3 | 49.0 | 55.0 | 37.3 | n/a | n/a | n/a |
| Aligned | 1 | 63.9 | 14.0 | 60.4 | 66.9 | 62.7 | 43.9 | 63.2 | 61.3 | 54.5 |
| Low-Rank | 1/16 | 54.2 | 9.4 | 53.0 | 66.9 | 62.3 | 43.7 | 60.2 | 58.3 | 51.0 |
| BitDelta | 1/16 | 54.8 | 10.6 | 51.8 | 64.2 | 62.6 | 41.6 | 60.9 | 60.3 | 50.9 |
| Delta-CoMe | 1/16 | **58.9** | **12.8** | **57.9** | **67.2** | **62.9** | **44.1** | **63.1** | **61.2** | **53.5** |

## 5.3 Results on More Backbone Models

To investigate the generalization abilities of the delta-compression methods, we conduct experiments on aligned models based on other representative backbone LLMs. For additional backbones, we utilize MISTRAL-7B-V0.1 (Jiang et al., 2023) and LLAMA-3-8B[7]. The corresponding aligned

---

[7] https://huggingface.co/meta-llama/Meta-Llama-3-8B.

Table 5: Results on other representative backbones. The backbone of OPENCHAT-3.5-0106 (Wang et al., 2023) is MISTRAL-7B-V0.1 (Jiang et al., 2023). Both MISTRAL-7B-V0.1 and LLAMA-3-8B are widely-used open-source LLMs.

| Method | $\alpha$ | OPENCHAT-3.5-0106 | | | | LLAMA-3-8B-INSTRUCT | | | | Ave. |
|---|---|---|---|---|---|---|---|---|---|---|
| | | GSM8K | HumanEval | TruthfulQA | SafetyBench | GSM8K | HumanEval | TruthfulQA | SafetyBench | |
| Backbone | 1 | 52.2 | 28.7 | 61.0 | 42.1 | 44.8 | 33.5 | 43.6 | 43.9 | 43.7 |
| Aligned | 1 | 77.1 | 73.2 | 78.4 | 61.0 | 78.5 | 61.6 | 68.2 | 51.6 | 68.7 |
| Low-Rank | 1/16 | 50.5 | 52.4 | 76.9 | 49.0 | 68.3 | 46.3 | 67.5 | 51.3 | 57.8 |
| BitDelta | 1/16 | 70.3 | 54.9 | 78.4 | 50.0 | 67.6 | 56.1 | 68.6 | 50.2 | 62.0 |
| Delta-CoMe | 1/16 | **74.8** | **59.8** | **78.9** | **62.6** | **77.1** | **60.4** | **69.1** | **51.8** | **66.8** |

models are OPENCHAT-3.5-0106 (Wang et al., 2023) and LLAMA-3-8B-INSTRUCT, respectively. As shown in Table 5, our proposed Delta-CoMe method maintains superior performance over the two baselines, demonstrating its generalization ability.

## 5.4 Delta-Compression vs. Delta-Tuning

A closely related area to delta-compression is delta-tuning. While delta-tuning primarily aims to reduce the training cost of LLMs, delta-compression focuses on reducing the storage and inference cost for multi-model serving. It remains unclear whether delta-compression outperforms delta-tuning when using the same delta size. To investigate this, we trained LoRA (Hu et al., 2022) modules for all model parameters to compare delta-compression with delta-tuning. We set the LoRA rank to 128 and the scale factor to 16, using a cosine warmup schedule with a warmup ratio of 0.04 and a peak learning rate of 1e-4. For each task, we trained the LoRA for 3 epochs. For mathematical LoRA, the training dataset is from Yu et al. (2023), which consists of 395K training examples. For code LoRA, the training set is from Wei et al. (2023), which contains 186K training examples. For a fair comparison, we fine-tune all model parameters using the same dataset as used for LoRA training. We then apply different delta-compression methods to both the fine-tuned mathematical and code LLMs.

Table 6 shows the results of both delta-tuning and delta-compression methods. The results reveal that LoRA achieves superior performance compared to the low-rank compression approach and BitDelta in the mathematical task. However, when it comes to the coding task, LoRA exhibits lower performance than both low-rank compression and BitDelta. By contrast, our proposed delta-compression method (i.e., Delta-CoMe) consistently outperforms LoRA across all four benchmarks. Specifically, the performance of our method is close to that of the uncompressed aligned

Table 6: Comparison between LoRA and delta-compression methods.

| Method | Math | | Code | | Ave. |
|---|---|---|---|---|---|
| | GSM8K | MATH | HumanEval | MBPP | |
| Backbone | 11.0 | 2.9 | 10.5 | 17.7 | 10.5 |
| Aligned | 65.4 | 18.6 | 43.2 | 44.9 | 43.0 |
| LoRA | 58.3 | 11.4 | 17.6 | 31.8 | 29.8 |
| Low-Rank | 54.8 | 5.5 | 26.2 | 42.6 | 32.3 |
| BitDelta | 47.8 | 10.7 | 26.2 | 41.9 | 31.7 |
| Delta-CoMe | **65.1** | **18.0** | **39.6** | **44.9** | **41.9** |

models (41.9 vs. 43.0), while the average score of LoRA is only 29.8. These results imply that learning an aligned model and then compressing it can achieve better results than delta-tuning.

## 5.5 Inference Speed and Memory Cost

For practical applications, we also examine the inference speed and memory cost of Delta-CoMe. In terms of inference speed, we implement a Triton kernel. Figure 3 shows the inference time of the PyTorch and Triton implementation of Delta-CoMe. Overall, we can achieve approximately a 3× speedup across different settings. As Figure 3a shows, we first conduct an ablation experiment on varying batch sizes. Our implemented Triton kernel is consistently faster than the PyTorch implementation with different batch size settings. As Figure 3b depicts, we conduct an ablation experiment on hidden size to verify the adaptability of the Triton kernel to models of different sizes.

The Triton kernel can maintain a substantial speedup across different hidden sizes, demonstrating its ability to adapt to various models.

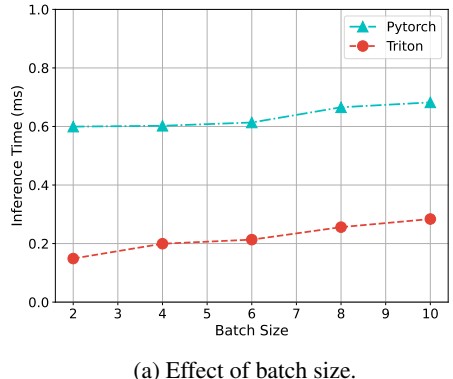

(a) Effect of batch size.

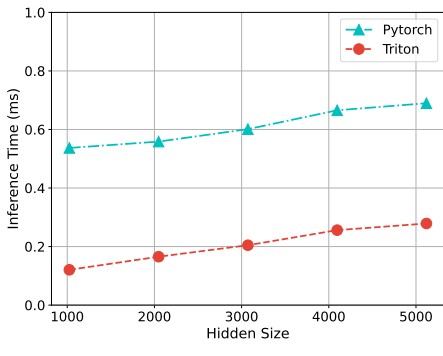

(b) Effect of hidden size.

Figure 3: Inference time of the PyTorch and Triton implementation of Delta-CoMe.

In Table 7, we show the GPU memory cost of deploying multiple aligned models that are fine-tuned from LLAMA-2-7B. The model parameters are represented in BF16 on a single 80G GPU. Without delta compression, a single GPU can not support 8 models, let alone more models. Using our proposed delta-compression method, we can load up to 50 models into one GPU, significantly reducing the deployment cost.

Table 7: GPU memory cost (GB).

| Num. of Models | w/o DC | w/ DC |
|----------------|--------|-------|
| 2 | 26.67 | 15.54 |
| 4 | 52.24 | 18.17 |
| 8 | OOM | 23.44 |
| 16 | OOM | 33.95 |
| 32 | OOM | 55.06 |
| 50 | OOM | 78.70 |

# 6  Analysis

## 6.1  Analysis of Quantization Error

To better understand the performance of various delta-compression methods, we estimate the quantization error as defined in Eq. (4). It is important to note that the error we calculate differs from that of GPTQ. Specifically, we use the mean square error between the activations of the uncompressed aligned model and those of the combination of the backbone model and the compressed delta model. The error is estimated on the GSM8K test set using WIZARDMATH-7B-V1.0 as the aligned model and LLAMA-2-7B as the backbone model. Since different layers have varying impacts on the final output (Wu et al., 2023), we distinguish low-, medium-, and high-layers when estimating the average quantization error. Specifically, the first 11 layers are designated as low-layers, the 12th to 22nd layers as medium-layers, and the last 10 layers as high-layers. Moreover, as outliers play a critical role in model compression (Dettmers et al., 2023; Lin et al., 2023), we also calculate the average error on outlier parameters. For each delta matrix $\Delta \mathbf{W}$, we select the top 1% of columns with the largest absolute values as outliers. Table 8 presents the results. We find that the average error of our methods (i.e., "Single" and "Triple") is substantially lower than both the low-rank baseline and BitDelta. Furthermore, the error of "Triple" is consistently less than that of "Single," reaffirming the necessity of mixed-precision compression for delta weights.

## 6.2  Case Study

We also present a detailed case study in Figure 4. Three delta-compression methods are examined: BitDelta, single-precision compression, and triple-precision compression. The reference answer is "104 hours". We observe that BitDelta makes mistakes initially, while single-precision compression generates an incorrect intermediate result at the second reasoning step. In contrast, our mixed-precision delta-compression method calculates the correct final answer.

Table 8: Approximation errors ($\times 10^{-2}$) at the activation level for different model parameters. "Low", "Medium", "High" represent low-, medium-, and high-layers, respectively. "All" means the error averaged across all the parameters, while "Out." denotes the average error estimated only on outliers.

| Param | Attn.Q_Proj | | | | | | Attn.K_Proj | | | | | |
|---|---|---|---|---|---|---|---|---|---|---|---|---|
| **Layer** | **Low** | | **Medium** | | **High** | | **Low** | | **Medium** | | **High** | |
| **Type** | **All** | **Out.** | **All** | **Out.** | **All** | **Out.** | **All** | **Out.** | **All** | **Out.** | **All** | **Out.** |
| Low-Rank | 0.75 | 2.24 | 4.24 | 14.31 | 4.47 | 10.28 | 0.87 | 9.90 | 4.79 | 34.04 | 4.82 | 31.41 |
| BitDelta | 0.97 | 2.48 | 4.66 | 14.48 | 4.84 | 10.01 | 1.09 | 10.34 | 5.16 | 33.03 | 5.14 | 28.06 |
| Single | 0.20 | 0.74 | 1.37 | 5.11 | 1.24 | 3.36 | 0.23 | 3.19 | 1.52 | 11.30 | 1.36 | 8.48 |
| Triple | **0.13** | **0.28** | **0.54** | **1.07** | **0.71** | **0.88** | **0.15** | **0.56** | **0.58** | **1.99** | **0.73** | **2.10** |

| Param | Attn.V_Proj | | | | | | Attn.O_Proj | | | | | |
|---|---|---|---|---|---|---|---|---|---|---|---|---|
| **Layer** | **Low** | | **Medium** | | **High** | | **Low** | | **Medium** | | **High** | |
| **Type** | **All** | **Out.** | **All** | **Out.** | **All** | **Out.** | **All** | **Out.** | **All** | **Out.** | **All** | **Out.** |
| Low-Rank | 0.41 | 3.61 | 1.84 | 8.27 | 2.93 | 4.64 | 0.01 | 0.13 | 0.10 | 0.39 | 0.38 | 5.94 |
| BitDelta | 0.45 | 3.60 | 1.95 | 8.02 | 3.18 | 4.85 | 0.01 | 0.13 | 0.11 | 0.44 | 0.37 | 5.45 |
| Single | 0.14 | 1.42 | 0.65 | 3.58 | 0.79 | 1.45 | 0.00 | 0.04 | 0.03 | 0.10 | 0.10 | 1.60 |
| Triple | **0.04** | **0.12** | **0.21** | **0.35** | **0.52** | **0.61** | 0.00 | **0.01** | **0.02** | **0.05** | **0.06** | **0.92** |

| Param | FFN.Up_Proj | | | | | | FFN.Gate_Proj | | | | | |
|---|---|---|---|---|---|---|---|---|---|---|---|---|
| **Layer** | **Low** | | **Medium** | | **High** | | **Low** | | **Medium** | | **High** | |
| **Type** | **All** | **Out.** | **All** | **Out.** | **All** | **Out.** | **All** | **Out.** | **All** | **Out.** | **All** | **Out.** |
| Low-Rank | 0.13 | 0.86 | 0.93 | 3.43 | 2.20 | 11.45 | 0.10 | 0.26 | 0.79 | 1.12 | 1.87 | 9.74 |
| BitDelta | 0.18 | 0.97 | 1.06 | 3.84 | 2.38 | 12.22 | 0.13 | 0.31 | 0.90 | 1.26 | 2.02 | 11.74 |
| Single | 0.03 | 0.17 | 0.27 | 1.08 | 0.56 | 3.10 | 0.02 | 0.06 | 0.23 | 0.35 | 0.47 | 2.14 |
| Triple | 0.03 | **0.11** | **0.15** | **0.49** | **0.39** | **2.01** | 0.02 | **0.03** | **0.14** | **0.15** | **0.35** | **1.64** |

Figure 4: Case study for different delta-compression methods, where only the triple-precision compression method proposed in this work can give the correct answer.

# 7 Conclusion

In this paper, we propose Delta-CoMe, a delta-compression method with mixed-precision inspired by the long-tail distribution of singular values in the delta weights. Delta-CoMe achieves near-lossless performance compared to uncompressed aligned models across various typical tasks, including math, code, chat, and multi-modal tasks. We validate the effectiveness of Delta-CoMe on several widely-used aligned LLMs, whose backbone pre-trained models include Llama-2, Llama-3, and Mistral. Experimental results demonstrate that Delta-CoMe outperforms several representative baselines by a considerable margin. We believe the newly introduced Delta-CoMe method has significant value for many practical applications, such as multi-tenant serving.

## Acknowledgments and Disclosure of Funding

This work is supported by the National Key R&D Program of China (No.2022ZD0116312), National Natural Science Foundation of China (No. 62236004, No. 62236011), and Institute Guo Qiang at Tsinghua University.

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

# A Limitation and Broader Impact

For limitations, on the one hand, we carried out extensive experiments to verify Delta-CoMe is near lossless in delta compression. However, we haven't explored mixed-precision in model compression. Recently, mixed-precision is applied widely in model compression and Delta-CoMe can provide a new perspective for model compression. On the other hand, our kernel is trivial, Wei et al. (2024) and Guo et al. (2024) have implemented more advanced kernels. We can draw on their methods to achieve higher acceleration ratios.

For broader impacts, this paper presents Delta-CoMe that mainly focuses on compression which can not only boost efficiency but also save GPU memory, may bring benefits to society.

# B Genetic Search for Bits Settings

In Section 5.2, we have elaborated on the setting of different bits. All our models employ the same configuration and have demonstrated near loss-less performance, which illustrates robustness.

Allocating different numbers to different bits (e.g. 16-bit, 8-bit) is a multi-objective optimization problem. We implemented a genetic algorithm to achieve a more fine-grained search. We use the following objective function,

$$f = \min \text{PPL}(x_1, x_2, x_3, x_4, x_5)$$

where $x_1, x_2, x_3, x_4, x_5$ indicating the number of 16-bit, 8-bit, 4-bit, 3-bit, 2-bit and PPL$(.)$ means we calculate perplexity using samples randomly chosen form C4 dataset. Table 9 illustrates the results, particularly in code tasks, where genetic search shows a significant improvement compared to greedy search. The average performance of genetic search across all tasks even surpasses that of the original half-precision models. However, the time and storage overhead of genetic search is much greater than that of greedy search.

Table 9: The performance of different bits allocate methods on 7B aligned models. "Greedy search" represents the method in Section 5.1.

| Method | $\alpha$ | WIZARDMATH | | MAGICODERS-CL | | LLAMA-2-CHAT | | LLAVA-V1.5 | | Ave. |
|---|---|---|---|---|---|---|---|---|---|---|
| | | GSM8K | MATH | HumanEval | MBPP | SafetyBench | TruthfulQA | GQA | TextVQA | |
| Backbone | 1 | 11.0 | 2.9 | 38.4 | 47.6 | 41.7 | 38.9 | n/a | n/a | n/a |
| Aligned | 1 | 55.2 | 10.9 | 70.7 | 69.2 | 59.5 | 44.6 | 62.0 | 58.2 | 53.5 |
| Greedy S. | 1/16 | 53.6 | 10.3 | 67.1 | 67.9 | 59.8 | 46.9 | 61.7 | 58.5 | 53.2 |
| Genetic S. | 1/16 | 53.6 | 10.3 | 69.5 | 68.9 | **59.9** | **47.3** | 61.7 | **58.5** | **53.7** |

# C Delta-CoMe Combine with Low-bit Backbone

Quantization methods (e.g., GPTQ, AWQ) have been widely used for quantizing backbones. It is of great significance for us to verify whether Delta-CoMe can still maintain good performance in low-bit backbone scenarios.

We evaluated the performance of Delta-CoMe using various backbones across multiple tasks in Table 10. We utilized GSM8K for math tasks, MBPP for code, TruthfulQA for chat, and TextVQA for multi-modal tasks. Table 10 has demonstrated that even when backbones are in low precision, Delta-CoMe can achieve performance similar to the original, indicating that Delta-CoMe can be further applied to backbones of various precision levels.

# D Exploring the boundary of Delta-CoMe

We have shown that Delta-CoMe can maintain near lossless performance under a 16× compression ratio. In the following, we attempt to explore the compression limits of Delta-CoMe. We employ

Table 10: Performance drop in 4-bit and 16-bit backbone across different tasks.

| Precision | Backbone | Tasks | Delta |
|---|---|---|---|
| 4-BIT BACKBONE | WizardMath 4-bit | 49.36 | n/a |
| | Llama2 4-bit + 1bit delta | 47.01 | -2.3 |
| 16-BIT BACKBONE | WizardMath 16-bit | 55.2 | n/a |
| | Llama2 16-bit + 1bit delta | 53.6 | -1.6 |
| 4-BIT BACKBONE | Magicoder 4-bit | 66.2 | n/a |
| | Codellama-python 4-bit + 1bit delta | 65.4 | -0.8 |
| 16-BIT BACKBONE | Magicoder 16-bit | 66.7 | n/a |
| | Codellama-python 16-bit + 1bit delta | 67.2 | +0.3 |
| 4-BIT BACKBONE | WizardMath 4-bit | 49.36 | n/a |
| | Llama2 4-bit + 1bit delta | 47.01 | -2.3 |
| 16-BIT BACKBONE | WizardMath 16-bit | 55.2 | n/a |
| | Llama2 16-bit + 1bit delta | 53.6 | -1.6 |
| 4-BIT BACKBONE | Llava-v1.5 4-bit | 57.68 | n/a |
| | Vicuna 4-bit + 1bit delta | 57.58 | -0.1 |
| 16-BIT BACKBONE | Llava-v1.5 16-bit | 58.2 | n/a |
| | Vicuna 16-bit + 1bit delta | 58.5 | +0.3 |

WizardMath-7B in GSM8K task to carry out our experiments which is shown in 11. For all the experiments, the rank share the same setting.

When the compression ratio is within 20×, Delta-CoMe still performs well. However, at a compression ratio of 32×, there is a noticeable decline in performance, but it still outperforms low-rank and low-bit methods, which only achieve a 16× compression ratio.

Table 11: Performance under different compression ratios for WizardMath-7B

| Model | w/o Comp. | 1/16 | 1/18 | 1/20 | 1/22 | 1/26 | 1/32 |
|---|---|---|---|---|---|---|---|
| WizardMath-7B | 55.2 | 53.6 | 52.2 | 51.9 | 51.2 | 50.1 | 48.8 |

