# OpenReview forum: "Delta-CoMe: Training-Free Delta-Compression with Mixed-Precision for Large Language Models"
_NeurIPS.cc/2024/Conference — NeurIPS 2024 poster_

### Official Review · Reviewer_ZM7A · 2024-07-06

**Soundness:** 2
**Presentation:** 3
**Contribution:** 2
**Rating:** 5
**Confidence:** 4

**Summary:**

This paper proposes an application of the mixed precision quantization technique to the singular vectors of the delta weights, which is encountered when serving multiple aligned LLMs.

**Strengths:**

Please see the “Questions” section.

**Weaknesses:**

Please see the “Questions” section.

**Questions:**

My review is as follows:

1) As a strength, this paper is well-written and makes it easy for the reader to follow the ideas.
2) The proposed method is a somewhat straightforward extension of mixed precision quantization to the singular vectors of the delta weights. The mixed precision quantization technique is well-known and commonly used by practitioners. I view the contribution of this paper as an extension of that. Because of that, I am not sure if the contributions of the paper could be considered novel. Having said that, it is good to see that the method performs well in multiple scenarios.
3) I understand that studying the compression of delta weights is important. In practice, the base model’s weights need to be quantized/compressed too for real-world situations. I don’t think the compression of base model weights and delta weights are completely orthogonal. I wonder what the authors think on the interdependency of the compression of base model weights and delta weights. In my opinion, it would improve the paper to include some results/discussion on this.
4) I’m wondering why the authors haven’t considered a more thorough search algorithm in Section 5.1 (or if they did, it would be great to include a discussion on it).

Minor:

5) Typo in Table 1: “Aligend”
6) Typo in Table 6: “HuamnEval”

**Limitations:**

Yes.

---

> ### Author Rebuttal · Authors · 2024-08-07
>
> Thanks for your questions.
>
> # For question 2.
> Representative works on mixed precision include Yao et al., Agile-Quant, and SpQR, which focus on applying mixed precision to model weights and activations, and Bablani et al. who apply mixed precision across different layers, while we adopt different precisions across different singular values in the feature space. Recently, Apple intelligence has also applied mixed precision across different layers. As the reviewer Wicf said, "While neither low-rank nor mixed precision quantization techniques are particularly novel, the combination of the two for delta-compression is novel and clever." As far as we know, we are the first to propose mixed precision compression in the feature space and in multi-modal settings.
>
> # For question 3
> We compress our backbone into 4-bit, and test their performance shown in the following table.  Our method can still retain performance though the backbone is compressed in low-bit format.
> |                   |                   | GSM8K  | delta  |
> |-------------------|-------------------|--------|--------|
> | 4-bit backbone    | WizardMath 4-bit  | 49.36% |  |
> |                   | Llama2 4-bit + 1bit delta | 47.01% |  -2.3%       |
> | 16-bit backbone   | WizardMath 16-bit | 55.2%  |  |
> |                   | Llama2 16-bit + 1bit delta | 53.6%  |  -1.6%       |
>
> |                   |                   | MBPP  | delta  |
> |-------------------|-------------------|--------|--------|
> | 4-bit backbone    | Magicoder 4-bit  | 66.2% |   |
> |                   | Codellama-python 4-bit + 1bit delta | 65.4% |    -0.8%    |
> | 16-bit backbone   | Magicoder 16-bit | 66.7%  |   |
> |                   | Codellama-python 16-bit+ 1bit delta | 67.2%  |    +0.3%    |
>
> |                   |                   | TruthfulQA  | delta  |
> |-------------------|-------------------|--------|--------|
> | 4-bit backbone    | WizardMath 4-bit  | 49.36% |  |
> |                   | Llama2 4-bit + 1bit delta | 47.01% |   -2.3%      |
> | 16-bit backbone   | WizardMath 16-bit | 55.2%  |   |
> |                   | Llama2 16-bit + 1bit delta | 53.6%  |  -1.6%      |
>
> |                   |                   | TextVQA  | delta  |
> |-------------------|-------------------|--------|--------|
> | 4-bit backbone    | Llava-v1.5 4-bit  | 57.68% |  |
> |                   | Vicuna 4-bit + 1bit delta | 57.58% |    -0.1%     |
> | 16-bit backbone   | Llava-v1.5 16-bit | 58.2%  |   |
> |                   | Vicuna 16-bit + 1bit delta| 58.5%  |  +0.3%      |
> # For question 4
> Due to limited pages, we did not include the detailed process of the decision of the number of different bit-widths in the current version. When setting $𝑟_{𝑏𝑒𝑔𝑖𝑛}$ and $𝑟_{𝑒𝑛𝑑}$, we decided based on minimizing the error between the activations of the compressed model and the original model. When searching for "Double Precision", using two 8-bit singular values results in the smallest error, while for "Triple Precision", 32 3-bit singular values results in the smallest error.
>
> #### Ablation on 8-bit in "Double Precision", changing the number of 8-bit from 1-8
>
> | Num. of 8-bit | 1   | 2    | 3    | 4    | 5    | 6    | 7    | 8    |
> |---------------|-----|------|------|------|------|------|------|------|
> | Error (× 10⁻²) | 0.85 | **0.81** | 0.84 | 0.86 | 0.88 | 0.85 | 0.95 | 0.94 |
> #### Ablation on 3-bit in "Triple Precision", changing the number of 3-bit from 8-64
>
> | Num. of 3-bit | 8   | 16   | 24   | 32   | 40   | 48   | 56   | 64   |
> |---------------|-----|------|------|------|------|------|------|------|
> | Error (× 10⁻²) | 0.77 | 0.77 | 0.76 | **0.74** | 0.75 | 0.76 | 0.78 | 0.77 |
>
> Based on the reviewers' insights, we regard the mixed precision issue as a multi-objective optimization problem, considering single precision to be a special case of mixed precision. We developed a genetic algorithm to address this problem, using the bit count of single precision as the initial solution. We use the following objective function, $f = min PPL(x1, x2, x3, x4, x5) $ where x1, x2, x3, x4, x5 indicating the number of 16-bit, 8-bit, 4-bit, 3-bit, 2-bit and $PPL(.)$ means we calculate perplexity in 128 samples randomly chosen form C4 dataset.  For each aligned model, we can automatically determine the mixing strategy through the genetic algorithm. The results demonstrate that the genetic algorithm can yield better results than greedy search, making our method easy to be applied to many different aligned models.
> | Models                | WizardMath      |                     | magicoder-S-CL   |             | Llama-2-7b-chat  |             | Llava-v1.5   |         |   Ave.
> |---------------|-----------------|---------------------|------------------|-------------|------------------|-------------|----------|--------|-----|
> | Tasks                 | GSM8K           | Math    | HumanEval        | Mbpp   | SafetyBench   | TruthfulQA    | GQA        | TextVQA   |
> | loss-based greedy search | 53.6        | 10.24    | 67.1            | 67.9       | 59.8     | 46.9       | 61.7            | 58.5       |    53.2
> | genetic search        | 53.6           | 10.24       | **69.5**        | **68.9**   | **59.9**   | **47.3**       | 61.7            | 58.5  |   **53.7**
>
> We also conducted experiments on the 13B model, where the genetic algorithm yielded better performance.
> | Models    | WizardMath      |         |
> |-------------|---------|-------------|
> | Tasks        | GSM8K   | Math    |
> | loss-based greedy search | 58.8   | 12.8    |
> | genetic search        | **59.4**       | **12.9**  |
>  The genetic algorithm demonstrated better performance, with the average performance on the 7B model even slightly surpassing "Aligned models". However, even with the same settings without genetic algorithm, our method's performance is close to that, indicating its generalization ability.

---

> ### Author Response · Authors · 2024-08-12
> **Invitation to Participate in the Discussion Period**
>
> Thank you very much for your insightful suggestions. We have provided detailed responses to your question. If you could participate in the discussion period, we would be very grateful.

---

> > ### Comment · Reviewer_ZM7A · 2024-08-12
> >
> > Thanks for the detailed answers to my questions. The new results are helpful. I'm however still on the fence about whether the novelty of this approach is substantial enough for a neurips paper.

---

### Official Review · Reviewer_PxGh · 2024-07-10

**Soundness:** 3
**Presentation:** 3
**Contribution:** 3
**Rating:** 6
**Confidence:** 4

**Summary:**

In the context of SVD compression of delta weight, the paper employs higher bitwidth for singular vectors corresponding to larger singular values. The available bitwidth 8, 3, 2 are empirically chosen. Once the bitwidths are assigned to the singular vectors, the vectors are group-wise quantized by GPTQ.

**Strengths:**

The paper is generally well-written and the experiments are very thorough. The combination of delta-compression and mixed-precision quantization appears to be novel. The method is straightforward and well-motivated.

**Weaknesses:**

Something is wrong with the latex encoding of this pdf. Sections are not detected properly by pdf readers, and there are odd reference errors such as the ones on line 26 and 30.

There are three steps of optimization being done: the number of largest singular vectors chosen, the bitwidth assigned to the singular vectors, and the quantization algoritm performed on the singular vectors. In the paper, the three steps are being done sequentially, leading to arbitrary, empirically-driven and possibly suboptimal decisions.

**Questions:**

Why is there an advantage to SVD the delta weights as opposed to SVD the aligned model weights directly?

**Limitations:**

Yes

---

> ### Author Rebuttal · Authors · 2024-08-07
>
> Thanks for your comment.
> # For weakness
> Due to limited pages, we did not include the detailed process of the decision of the number of different bit-widths in the current version. When setting $𝑟_{𝑏𝑒𝑔𝑖𝑛}$ and $𝑟_{𝑒𝑛𝑑}$, we decided based on minimizing the error between the activations of the compressed model and the original model. When searching for "Double Precision", using two 8-bit singular values results in the smallest error, while for "Triple Precision", 32 3-bit singular values results in the smallest error.
>
> #### Ablation on 8-bit in "Double Precision", changing the number of 8-bit from 1-8
>
> | Num. of 8-bit | 1   | 2    | 3    | 4    | 5    | 6    | 7    | 8    |
> |---------------|-----|------|------|------|------|------|------|------|
> | Error (× 10⁻²) | 0.85 | **0.81** | 0.84 | 0.86 | 0.88 | 0.85 | 0.95 | 0.94 |
> #### Ablation on 3-bit in "Triple Precision", changing the number of 3-bit from 8-64
>
> | Num. of 3-bit | 8   | 16   | 24   | 32   | 40   | 48   | 56   | 64   |
> |---------------|-----|------|------|------|------|------|------|------|
> | Error (× 10⁻²) | 0.77 | 0.77 | 0.76 | **0.74** | 0.75 | 0.76 | 0.78 | 0.77 |
>
> Based on the reviewers' insights, we regard the mixed precision issue as a multi-objective optimization problem, considering single precision to be a special case of mixed precision. We developed a genetic algorithm to address this problem, using the bit count of single precision as the initial solution. We use the following objective function, $f = min PPL(x1, x2, x3, x4, x5) $ where x1, x2, x3, x4, x5 indicating the number of 16-bit, 8-bit, 4-bit, 3-bit, 2-bit and $PPL(.)$ means we calculate perplexity in 128 samples randomly chosen form C4 dataset.  For each aligned model, we can automatically determine the mixing strategy through the genetic algorithm. The results demonstrate that the genetic algorithm can yield better results than greedy search, making our method easy to be applied to many different aligned models.
> | Models                | WizardMath      |                     | magicoder-S-CL   |             | Llama-2-7b-chat  |             | Llava-v1.5   |         |   Ave.
> |---------------|-----------------|---------------------|------------------|-------------|------------------|-------------|----------|--------|-----|
> | Tasks                 | GSM8K           | Math    | HumanEval        | Mbpp   | SafetyBench   | TruthfulQA    | GQA        | TextVQA   |
> | loss-based greedy search | 53.6        | 10.24    | 67.1            | 67.9       | 59.8     | 46.9       | 61.7            | 58.5       |    53.2
> | genetic search        | 53.6           | 10.24       | **69.5**        | **68.9**   | **59.9**   | **47.3**       | 61.7            | 58.5  |   **53.7**
>
> We also conducted experiments on the 13B model, where the genetic algorithm yielded better performance.
> | Models    | WizardMath      |         |
> |-------------|---------|-------------|
> | Tasks        | GSM8K   | Math    |
> | loss-based greedy search | 58.8   | 12.8    |
> | genetic search        | **59.4**       | **12.9**  |
>  The genetic algorithm demonstrated better performance, with the average performance on the 7B model even slightly surpassing "Aligned models". However, even with the same settings without genetic algorithm, our method's performance is close to that, indicating its generalization ability.
>
> # For question
> 1. Compress aligned models directly using SVD can easily achieve bad performance in low-bit setting. As the following table using SVD compress the model into 1-bit or 2-bit settings resulting in a performance drop to 0 in math and code tasks.
> | Tasks         | GSM8K | Math | HumanEval | Mbpp |
> |---------------|-------|------|-----------|------|
> | Compress 1-bit| 0     | 0    | 0         | 0    |
> | Compress 2-bit| 0     | 0    | 0         | 0    |
> 2. Recently Onebit (Xu et al.) find compress aligned model needs more sophisticated algorithms and  post-training is also needed.  Even in this carefully designed setting, there is still a significant performance drop. For example,  in Table 2 of OneBit, the average performance on 6 benchmarks decreased from 64.06 to 51.33 and from 66.39 to 55.17 on Llama-2-7B and Llama-2-13B, respectively.
> 3. In our experiments, delta exhibits a long-tail distribution characteristic, and using SVD compression algorithms can retain the model in a near-lossless manner, compressing it to an equivalent of 1-bit. Therefore, compressing delta-weights can achieve better performance.

---

> ### Author Response · Authors · 2024-08-12
> **Invitation to Participate in the Discussion Period**
>
> Thank you very much for your insightful suggestions. We have provided detailed responses to your question. If you could participate in the discussion period, we would be very grateful.

---

> > ### Comment · Reviewer_PxGh · 2024-08-12
> >
> > Dear Authors, My concerns have been addressed accordingly. I have raised my score to 6 and confidence to 4.

---

### Official Review · Reviewer_Wicf · 2024-07-12

**Soundness:** 4
**Presentation:** 3
**Contribution:** 3
**Rating:** 7
**Confidence:** 3

**Summary:**

This paper proposes an improved way to apply delta-compression for aligned language models (compact representations of the difference in weights between the pretrained and finetuned language models), which is just as effective w.r.t compression as the most extreme existing binary quanitzation strategy (BitDelta), but resolving many existing quality issues w.r.t. math and code tasks over previous methods. The key insight is that it is possible to combine the best of both worlds between low-rank methods (selecting the top singular values) and quantization methods (e.g. BitDelta), and represent more singular values by using decreasing precision for decreasing singular values: a mixed precision technique overall. The authors call this technique Delta-CoMe. When comparing low-rank, BitDelta, and Delta-CoMe at a fixed compression rate (equal to the binarization / BitDelta rate, which is quite aggressive), Delta-CoMe consistently outperforms all baselines and is close to matching performance with the full precision aligned model. Experiments take place over a diverse set of tasks (math, code, QA, safety, multimodal) as well as a diverse set of models at 7B and 13B scales and aligned versions (Llama 2, WizardMath, MagicCoders, Llama-2-Chat, LLAVA v1.5, OpenChat, Llama-3-8b-Instruct). The authors also provide quality comparisons between delta-compression methods and delta-tuning (i.e. LoRA), demonstrating that Delta-CoMe outperforms LoRA w.r.t quality.

**Strengths:**

1. While neither low-rank or mixed precision quantization techniques are particularly novel, the combination of the two for delta-compression is novel and clever in the way that the framework smoothly models the trade-off between representing everything with low (binary) precision, and representing the most important features with full precision. Although this is not explored deeply in this paper (the authors recognize that this is a proof of concept of the effectiveness rather than an optimized solution), this tradeoff could be tuned to suit a variety of cases. This idea could also potentially be applied in other areas, such as PEFT or general mixed-precision work, as previous works used quite different heuristics to select which weights to have mixed precision on.

2. The results presented are significant. The approach significantly outperforms the baselines in all aspects (math, code, chat, multimodal), with the biggest improvements in math and code. The approach is the only one to come close to matching the non-compressed aligned model overall over all capabilites. This will be of interest to anyone interested in delta-compression literature or even PEFT techniques. It is the first work to apply mixed-precision to delta weights.

3. The experiments was done on a wide variety of models (math specialized tunings, code specialized tunings, chat tunings, multimodal tunings) and settings (7/8B, 13B, over base models like Llama2, Llama3 and Mistral) enough to demonstrate confidence in the ability of the technique. The authors also provide nice additional analysis of quantization error, and comparison in quality against vanilla LoRA as additional evidence for their approach.

4. The paper is overall well structured and well written, which makes the problem easy to understand and the author's intuition and investigation relatively easy to follow. There are some improvements that could be made (below), but it is nice overall.

**Weaknesses:**

1. In Table 4, it does seem like on the larger 13B models WizardMath models, the performance recovery in GSM8k and MATH is not as strong as on 7 or 8B scale models. It does bring into question whether there are limitations to this approach not discussed in this work that may limit general adoptability. Whether that means not working as well on larger scales, or for the certain type of post-training happening in WizardMath.

1. Although I understand the author's motivation of using greedy decoding to encourage reproducibility and decrease noise in the evaluations, many models especially on chat tasks use sampling based-decoding, and having results here would improve the overall robustness of the experiments to verify that the techniques still work well.

1. Presentation a couple points in the paper could be improved :

    a. While the authors do note that the greedy search strategy to allocate mixed-precision to the singular values presented in Section 5.1 represents a proof of concept to show that even such an unoptimized approach can achieve good results, it would be valuable to understand how some of the design choices here were informed. For example, in the 3-precision setting, the range of singular vectors for the 2nd precision is set to be between 2 and 34, which feels arbitrary. What is the intuition behind this or were there ablations ran here? Being that Section 5.1 is the core description of the mixed-precision strategy, more detail and clarity here would be appreciated.

    b. (Minor) The introduction could be cleaned up a bit, with lines 21-36 being oddly detailed. The claim on line 44-46 to motivate the work is also not super well founded -- there are models on LMSys showing general ability with ~20B parameters, not too much bigger than those considered in this work.

**Questions:**

1. What is the intuition behind choosing the singular vector bands for the different precisions the way that you do in Section 5.1?

1. Do you think the ideas here could apply to mixed-precision approaches more generally?

1. Why is the approach named Delta-CoMe? It appears to stand for delta compression method which seems very generic? And has nothing to do with the details of the approach.

1. Are there any practical limitations e.g. hardware with mixed-precision delta-compression not covered in this paper?

**Limitations:**

The authors do clearly point out that a major limitation of the work is the lack of optimization for how the mixed precision is assigned. This makes the approach more of a proof of concept that even a naive solution works, rather than an optimized method. This is perfectly fine.

It would be valuable to understand though whether theres actually limitations in deploying this type of approach on actual hardware with multi-tenant hosting.

It would also be interesting to understand the boundaries to when this approach works or does not work, e.g. at what compression ratio would it break, whether theres a model scale too small or too large, etc.

---

> ### Author Rebuttal · Authors · 2024-08-07
>
> Thanks for your questions, weaknesses, and limitations which can bring much improvement to our paper.
>
> # For question 1.
> Due to limited pages, we did not include the detailed process of the decision of the number of different bit-widths in the current version. When setting $𝑟_{𝑏𝑒𝑔𝑖𝑛}$ and $𝑟_{𝑒𝑛𝑑}$, we decided based on minimizing the error between the activations of the compressed model and the original model.
> #### Ablation on 8-bit in "Double Precision", changing the number of 8-bit from 1-8
>
> | Num. of 8-bit | 1 | 2 | 3  | 4  | 5  | 6  | 7  | 8  |
> |------|-----|----|----|----|---|----|----|----|
> | Error (× 10⁻²) | 0.85 | **0.81** | 0.84 | 0.86 | 0.88 | 0.85 | 0.95 | 0.94 |
> #### Ablation on 3-bit in "Triple Precision", changing the number of 3-bit from 8-64
>
> | Num. of 3-bit | 8  | 16 | 24 | 32  | 40  | 48 | 56  | 64  |
> |--------|-----|------|------|------|------|------|------|------|
> | Error (× 10⁻²) | 0.77 | 0.77 | 0.76 | **0.74** | 0.75 | 0.76 | 0.78 | 0.77 |
> # For question 2.
> Based on the reviewers' insights, we regard the mixed precision issue as a multi-objective optimization problem, considering single precision to be a special case of mixed precision. We developed a genetic algorithm to address this problem, using the bit count of single precision as the initial solution. We use the following objective function, $f = min PPL(x1, x2, x3, x4, x5) $ where x1, x2, x3, x4, x5 indicating the number of 16-bit, 8-bit, 4-bit, 3-bit, 2-bit and $PPL(.)$ means we calculate perplexity in 128 samples randomly chosen form C4 dataset.  For each aligned model, we can automatically determine the mixing strategy through the genetic algorithm. The results demonstrate that the genetic algorithm can yield better results than greedy search, making our method easy to be applied to many different aligned models.
> | Models     | WizardMath      |           | magicoder-S-CL   |      | Llama-2-7b-chat  |   | Llava-v1.5   |         |   Ave.
> |----------|---------|------------|-------|--------|-------------|--------|-----|--------|-----|
> | Tasks       | GSM8K    | Math    | HumanEval        | Mbpp   | SafetyBench   | TruthfulQA    | GQA        | TextVQA   |
> | loss-based greedy search | 53.6   | 10.24    | 67.1     | 67.9  | 59.8     | 46.9       | 61.7    | 58.5       |    53.2
> | genetic search        | 53.6      | 10.24   | **69.5**   | **68.9**   | **59.9**   | **47.3**   | 61.7    | 58.5  |   **53.7**
>
> We also conducted experiments on the 13B model, where the genetic algorithm yielded better performance.
> | Models    | WizardMath   |    |
> |-------------|---------|-------------|
> | Tasks    | GSM8K   | Math    |
> | loss-based greedy search | 58.8   | 12.8    |
> | genetic search        | **59.4**   | **12.9**  |
>
>  The genetic algorithm demonstrated better performance, with the average performance on the 7B model even slightly surpassing "Aligned models". However, even with the same settings without genetic algorithm, our method's performance is close to that, indicating its generalization ability.
>
> Recently, mixed precision is being more widely used such as Apple intelligence (Gunter, Tom, et al. ) allocate different bits to different layers which achieves 3.5-bit on average and Jinhao Li et al. employ mixing 2-bit, 4-bit and 16-bit that ultimately attains an average of 2-bit. Recently, mixed precision has received more attention than before, and the use of mixed precision methods has also been widely applied to model compression. We believe that our method could also be further applied to model compression, which is the direction of our next exploration.
> # For question 3.
>  Thanks for your great question. We will give a more proper title.
> # For question 4.
>   Ablation on batch, we set sequence length to 128, our method achieves 3x speedup than Pytorch. Further, Han Guo et al. and Jianyu et al. have implemented more advanced kernels. We can draw on their methods to achieve higher acceleration ratios.
>  #### Ablation on batch, we set sequence length to 128, our method achieves 3x speedup than Pytorch.
> | bsz | Linear (ms) | Our (ms) |
> |-----|-------------|----------|
> | 2   | 0.5995      | 0.1488   |
> | 4   | 0.6024      | 0.1995   |
> | 6   | 0.6136      | 0.2134   |
> | 8   | 0.6656      | 0.2561   |
>  #### Ablation on hidden_size, we set batch 8.
> | hidden_size | Linear (ms) | Our (ms) |
> |-------------|--------------|----------|
> | 1024        | 0.5369       | 0.1206   |
> | 2048        | 0.5586       | 0.1654   |
> | 3072        | 0.6011       | 0.2046   |
> | 4096        | 0.6656       | 0.2561   |
> | 5120        | 0.6897       | 0.2788   |
>
> # For limitation
>   Thanks for your great insight, we conducted a more granular exploration, compressing the model up to 32×. We use WizardMath-7B in GSM8K task and results are shown in the following table. When the compression ratio is within 20×, our method still performs well. However, at a compression ratio of 32×, there is a noticeable decline in performance, but it still outperforms low-rank and low-bit methods, which only achieve a 16× compression ratio.
> |                | w/o Comp.    | 1/16    | 1/18    | 1/20    | 1/22    | 1/26    | 1/32 |
> |----------------|---------|---------|---------|---------|---------|---------|-----------|
> | WizardMath-7B  | 55.2  | 53.6  | 52.2  | 51.9  | 51.2  | 50.1   | 48.8     |
>
> # For weakness
> We have used sample decoding algorithm and set decoding temperature = 0.2.  We run 5 times and calculate the mean value and confidence interval shown in the table.
> |          | GSM8K           | truthfulQA      | MBPP            | TextVQA         |
> |----------|------------------|-----------------|-----------------|-----------------|
> | Low-rank | 42.6 (41.9, 43.3)| 42.2 (42.0, 42.4)| 65.4 (65.1, 65.9)| 52.9 (52.5, 53.6)|
> | Bitdelta | 45.2 (44.0, 46.4)| 40.8 (40.4, 41.3)| 65.7 (65.4, 65.9)| 56.5 (56.2, 57.1)|
> | Ours     | 53.8 (53.0, 54.6)| 46.9 (46.3, 47.6)| 68.0 (67.6, 68.5)| 58.3 (57.9, 58.9)|

---

> > ### Author Response · Authors · 2024-08-08
> > **Performance on hardwares**
> >
> > To accelerate the inference of our proposed method, we implemented a computation kernel using Triton, which integrates the de-quantization process with matrix multiplication. This kernel supports the multiplication of matrices with different bit-widths.
> > By using our customized Triton kernal, we can significantly improve the forward speed of Delta-CoMe:
> >
> > Ablation on batch, we set sequence length to 128, our method achieves 3x speedup than Pytorch.
> > | bsz | Delta-CoMe w/ PyTorch (ms) | Our Customized Triton Kernal (ms)|
> > |-----|-------------|----------|
> > | 2   | 0.5995      | 0.1488   |
> > | 4   | 0.6024      | 0.1995   |
> > | 6   | 0.6136      | 0.2134   |
> > | 8   | 0.6656      | 0.2561   |
> >
> > Ablation on hidden_size, we set batch 8.
> >
> > | hidden_size | Delta-CoMe w/ PyTorch (ms) | Our Customized Triton Kernal (ms) |
> > |-------------|--------------|----------|
> > | 1024        | 0.5369       | 0.1206   |
> > | 2048        | 0.5586       | 0.1654   |
> > | 3072        | 0.6011       | 0.2046   |
> > | 4096        | 0.6656       | 0.2561   |
> > | 5120        | 0.6897       | 0.2788   |
> > Further, Han Guo et al. and Jianyu et al. have implemented more advanced kernels. We can draw on their methods to achieve higher acceleration ratios.
> >
> > To further demonstrate the hardware advantages of our method, in addition to inference speed, we have also provided memory usage in the table below.
> > | Num. deployed models | Original Storage | Our Storage |
> > |-----------------|------------------|-------------|
> > | 2               | 26.67G           | 15.39G      |
> > | 4               | 52.24G           | 17.02G      |
> > | 8               | OOM              | 20.28G      |
> > | 16              | OOM              | 26.79G      |
> > | 32              | OOM              | 39.84G      |
> > | 64              | OOM              | 66.68G      |
> > Our method can save GPU memory significantly.

---

> > > ### Comment · Reviewer_Wicf · 2024-08-12
> > > **Response**
> > >
> > > Thank you authors for providing the wealth of additional results. I strongly urge the authors to update the manuscript in future versions or camera ready with these additional results, as I do really think that the work greatly benefits from it. With this being said I would like to keep my score at this time -- which genuinely represents a well executed and empirically sound work that is valuable and I believe should be accepted to the conference. Only limitations from a higher score would be regarding the scope and size of the contribution of the work.

---

### Official Review · Reviewer_vBME · 2024-07-21

**Soundness:** 2
**Presentation:** 3
**Contribution:** 2
**Rating:** 6
**Confidence:** 4

**Summary:**

This paper introduces Delta-CoMe, a delta compression method for LLM. It proposes to combine low-rank compression and low-bit compression together to achieve better performance. Specifically, it applies mixed-precision quantization for different singular vectors based on their singular values. The experimental results demonstrate good performance of the proposed method.

**Strengths:**

1. The proposed method is easy and straight to implement for other LLMs, based on existing open-source tools.
2. The motivation and the proposed idea is clear and supported by the experiment results.

**Weaknesses:**

1. The bit-width in mixed-precision seems to be a multi-objective optimization problem, and the authors adopt a greedy search in this paper. In Table 2, different settings vary a lot in performance. This suggests that it requires to decide the bit-width carefully, making the method not easy to generalize well to other tasks/models. Performance may not be guaranteed before doing a search.
2. The value of $r_{begin}$ and $r_{end}$ seem to be set intentionally and empirically without explanation or study. This would also impact the generalization of the method.

**Questions:**

1. In equation 5, why the quantization of $V$ is $Quant_k(V, X)$ without $U$ and $\Sigma$ as in the quantization of $U$?
2. In Section 5.1, how are $r_{begin}$ and $r_{end}$ set to those values?

---

> ### Author Rebuttal · Authors · 2024-08-07
>
> Thanks for your questions
> # For question 1
> In delta-compression, we consider compressing $U$ and $V^{T}$. For $U$ and $V^{T}$, their inputs are crucial for adjusting weights during the quantization process, as illustrated in GPTQ.  In a forward pass $Y = W X + (UΣV^{T}) X$, the input of $V^{T}$ is $X$, while the input of $U$ is $ΣV^{T}X$. Therefore, the quantization of $V$ is $Quant(V, X)$, while the quantization of $U$ is $Quant(U, ΣV^{T}X)$.
> # For question 2
> Due to limited pages, we did not include the detailed process of the decision of the number of different bit-widths in the current version. When setting $𝑟_{𝑏𝑒𝑔𝑖𝑛}$ and $𝑟_{𝑒𝑛𝑑}$, we decided based on minimizing the error between the activations of the compressed model and the original model. When searching for "Double Precision", using two 8-bit singular values results in the smallest error, while for "Triple Precision", 32 3-bit singular values results in the smallest error.
>
> #### Ablation on 8-bit in "Double Precision", changing the number of 8-bit from 1-8
>
> | Num. of 8-bit | 1   | 2    | 3    | 4    | 5    | 6    | 7    | 8    |
> |---------------|-----|------|------|------|------|------|------|------|
> | Error (× 10⁻²) | 0.85 | **0.81** | 0.84 | 0.86 | 0.88 | 0.85 | 0.95 | 0.94 |
> #### Ablation on 3-bit in "Triple Precision", changing the number of 3-bit from 8-64
>
> | Num. of 3-bit | 8   | 16   | 24   | 32   | 40   | 48   | 56   | 64   |
> |---------------|-----|------|------|------|------|------|------|------|
> | Error (× 10⁻²) | 0.77 | 0.77 | 0.76 | **0.74** | 0.75 | 0.76 | 0.78 | 0.77 |
>
> Based on the reviewers' insights, we regard the mixed precision issue as a multi-objective optimization problem, considering single precision to be a special case of mixed precision. We developed a genetic algorithm to address this problem, using the bit count of single precision as the initial solution. We use the following objective function, $f = min PPL(x1, x2, x3, x4, x5) $ where x1, x2, x3, x4, x5 indicating the number of 16-bit, 8-bit, 4-bit, 3-bit, 2-bit and $PPL(.)$ means we calculate perplexity in 128 samples randomly chosen form C4 dataset.  For each aligned model, we can automatically determine the mixing strategy through the genetic algorithm. The results demonstrate that the genetic algorithm can yield better results than greedy search, making our method easy to be applied to many different aligned models.
> | Models                | WizardMath      |                     | magicoder-S-CL   |             | Llama-2-7b-chat  |             | Llava-v1.5   |         |   Ave.
> |---------------|-----------------|---------------------|------------------|-------------|------------------|-------------|----------|--------|-----|
> | Tasks                 | GSM8K           | Math    | HumanEval        | Mbpp   | SafetyBench   | TruthfulQA    | GQA        | TextVQA   |
> | loss-based greedy search | 53.6        | 10.24    | 67.1            | 67.9       | 59.8     | 46.9       | 61.7            | 58.5       |    53.2
> | genetic search        | 53.6           | 10.24       | **69.5**        | **68.9**   | **59.9**   | **47.3**       | 61.7            | 58.5  |   **53.7**
>
> We also conducted experiments on the 13B model, where the genetic algorithm yielded better performance.
> | Models    | WizardMath      |         |
> |-------------|---------|-------------|
> | Tasks        | GSM8K   | Math    |
> | loss-based greedy search | 58.8   | 12.8    |
> | genetic search        | **59.4**       | **12.9**  |
>  The genetic algorithm demonstrated better performance, with the average performance on the 7B model even slightly surpassing "Aligned models". However, even with the same settings without genetic algorithm, our method's performance is close to that, indicating its generalization ability.

---

> ### Author Response · Authors · 2024-08-13
> **Invitation to Join the Discussion Period**
>
> Thank you very much for your insightful suggestions. We have provided detailed responses to your question. If you have any other questions, we sincerely invite you to participate in the discussion period if possible. Thanks very much!

---

### Author Rebuttal · Authors · 2024-08-07

Thank all the reviewers for the constructive suggestions. We will take into account the advice to improve the manuscript!

---

### Decision · Program_Chairs · 2024-09-25

**Decision:**

Accept (poster)

**Comment:**

This paper presets a novel approach to apply delta-compression for aligned language models (from pretrained and finetued language models). The proposed approach try to combine the existing low-rank and low-bit compression methods together to achieve better performance. More specially, the approach applies mixed precision quantization for different SV based on their values. All reviews agree the paper is novel and the experiments indeed show the merit of the proposed approach.